# Accurate contact-based modelling of repeat proteins predicts the structure of new repeats protein families

**Claudio Bassot**[1,2], **Arne Elofsson**[1,2]*

**1** Science for Life Laboratory, Solna, Sweden, **2** Dep. of Biochemistry and Biophysics, Stockholm University, Stockholm, Sweden

* arne@bioinfo.se

**Data Availability Statement:** All the protein models, contact prediction, and Multiple Sequence alignments are available at https://figshare.com/articles/dataset/Repeats_Proteins_contact_prediction_based_modelling_datasets/9995618.

## Abstract

Repeat proteins are abundant in eukaryotic proteomes. They are involved in many eukaryotic specific functions, including signalling. For many of these proteins, the structure is not known, as they are difficult to crystallise. Today, using direct coupling analysis and deep learning it is often possible to predict a protein's structure. However, the unique sequence features present in repeat proteins have been a challenge to use direct coupling analysis for predicting contacts. Here, we show that deep learning-based methods (trRosetta, DeepMetaPsicov (DMP) and PconsC4) overcomes this problem and can predict intra- and inter-unit contacts in repeat proteins. In a benchmark dataset of 815 repeat proteins, about 90% can be correctly modelled. Further, among 48 PFAM families lacking a protein structure, we produce models of forty-one families with estimated high accuracy.

## Author summary

Repeat proteins are widespread among organisms and particularly abundant in eukaryotic proteomes. Their primary sequence presents repetition in the amino acid sequences that origin structures with repeated folds/domains. Although the repeated units often can be recognised from the sequence alone, often structural information is missing. Here, we used contact prediction for predicting the structure of repeats protein directly from their primary sequences. We benchmark the methods on a dataset comprehensive of all the known repeated structures. We evaluate the contact predictions and the obtained models for different classes of repeat proteins. Further, we develop and benchmark a quality assessment (QA) method specific for repeat proteins. Finally, we used the prediction pipeline for all PFAM repeat families without resolved structures and found that forty-one of them could be modelled with high accuracy.

## Introduction

Repeat proteins contain periodic units in the primary sequence that are likely the result of duplication events at the genetic level [1]. Repeat proteins emerge through replication slippage

**Funding:** This project has received funding from the European Union's Horizon 2020 research and innovation programme under the Marie Skłodowska-Curie grant agreement No 823886. AE is funded by grants from the Swedish Natural Science Research Council (Vetenskapsrådet) No VR-NT 2016-03798. SNIC provided computational resources under grant agreement No SNIC 2020/5-300. The funders had no role in study design, data collection and analysis, decision to publish, or preparation of the manuscript.

**Competing interests:** The authors have declared that no competing interests exist

[2] and double-strand break repair [3]. This protein class is present in all genomes but is more frequent in eukaryotic organisms [4–6] where they are involved in a wide range of functions [7]. In particular, due to their extended structures, repeat proteins often behave as molecular scaffolds in protein signalling or for protein complexes as WD40 domain [8], or ankyrin repeats [9,10]. Repeat proteins are usually conserved among orthologs [4,11] while exhibiting a more accelerated evolution and divergence among paralogs [11].

A classification of repeat proteins was proposed by Kajava [12,13] based on the length of the repeat units and the tertiary structure of the repeat units. According to Kajava's classification, there are five classes of repeat proteins. However, in this study, we ignore class I and II because there are no available structures for class I, and class II structures are folded in a coiled-coil structure possible to predict using other methods. Moreover, the extreme amino acid compositional bias of many of these proteins makes it difficult to identify the coevolving residues in these two classes.

The dataset used in our study contains three classes of proteins divided into 20 subclasses by their secondary structure, according to RepeatsDB [14], Fig 1. The three types are; class III extended repeats (e.g. α and β solenoids); class IV closed repeats structures (e.g. TIM and β barrels and β-propeller), and class V where the units appear as separate domains on a string. Further, class V the repeat units are longer than in the other classes.

The solenoid structures (subclasses III.1, III.2 III.3) dominate Class III [13], and these proteins contain a wide range of repeated units (from 4 to 38), Fig 1. The length of the individual unit is also quite variable (from 10 to 50 residues) [14], with β-solenoids having significantly shorter repeats compared with α and α/β solenoid [13].

Members of class IV are constrained in variability by the closed fold. Indeed, despite ten subclasses of different units, the number of units varies between 3 and 16, and proteins with more than ten repeat units are rare. Even in this class, the length of the repeats units varies between 10 to 50 residues [13]. Finally, class V proteins are made up of the extended repeat units, often longer than 40 residues [14]. Each unit folds into proper domains, and they only have few inter-unit contacts.

Many repeat protein families lack a resolved structure. For these protein families, residue-residue contact prediction is the best method to obtain structural information [15]. Contact prediction methods use residue-residue co-evolution from multiple sequence alignment and identify the residues' evolutionary constraints imposed by the tertiary protein structure [16]. Nevertheless, repeat proteins are a difficult target for contact prediction; the internal symmetry introduces artefacts in the contact map at a distance corresponding to the repeated units [17].

Here, we benchmark the deep-learning-based contacts prediction programs PconsC4 [18] trRosetta [19], DeepMetaPsicov [20] against the GaussDCA [21] on a comprehensive dataset generated from RepeatsDB [14]. The predicted contacts were then used as constraints to generate protein models. The model quality was evaluated, combining the quality assessment scores from Pcons [22] and QmeanDisCo [23] through a random forest regression. Based on the benchmark, we propose models for the protein structures of PFAM protein families missing resolved structures.

## Results and discussion

### General contact prediction analysis in repeat proteins

To assess the quality of the contacts predictions among repeat protein classes, we generate a dataset of proteins using the reviewed entries of RepeatsDB [14] and then clustered at 40% sequence identity. For each repeats region in the dataset, we also extracted a representative

## CLASS III

III.1 β-solenoid

III.2 α/β solenoid

III.3 α-solenoid

III.4  β trefoil / β hairpins

III.5  anti-parallel β layer / β hairpins

## CLASS IV

IV.1 TIM-barrel

IV.2 β-barrel / β hairpins

IV.3 β-trefoil

IV.4 β-propeller

IV.5 α/β prism

IV.6 α-barrel

IV.7 α/β barrel

IV.8 α/β propeller

IV.9 α/β trefoil

IV.10 aligned prism

## CLASS V

V.1 α-beads

V.2 β-beads

V.3 α/β-beads

V.4 β sandwich beads

V.5 α/β sandwich beads

**Fig 1. Repeats proteins classification.** Representation of the repeats classes and subclasses as classified in repeatsDB 2.0 [14].

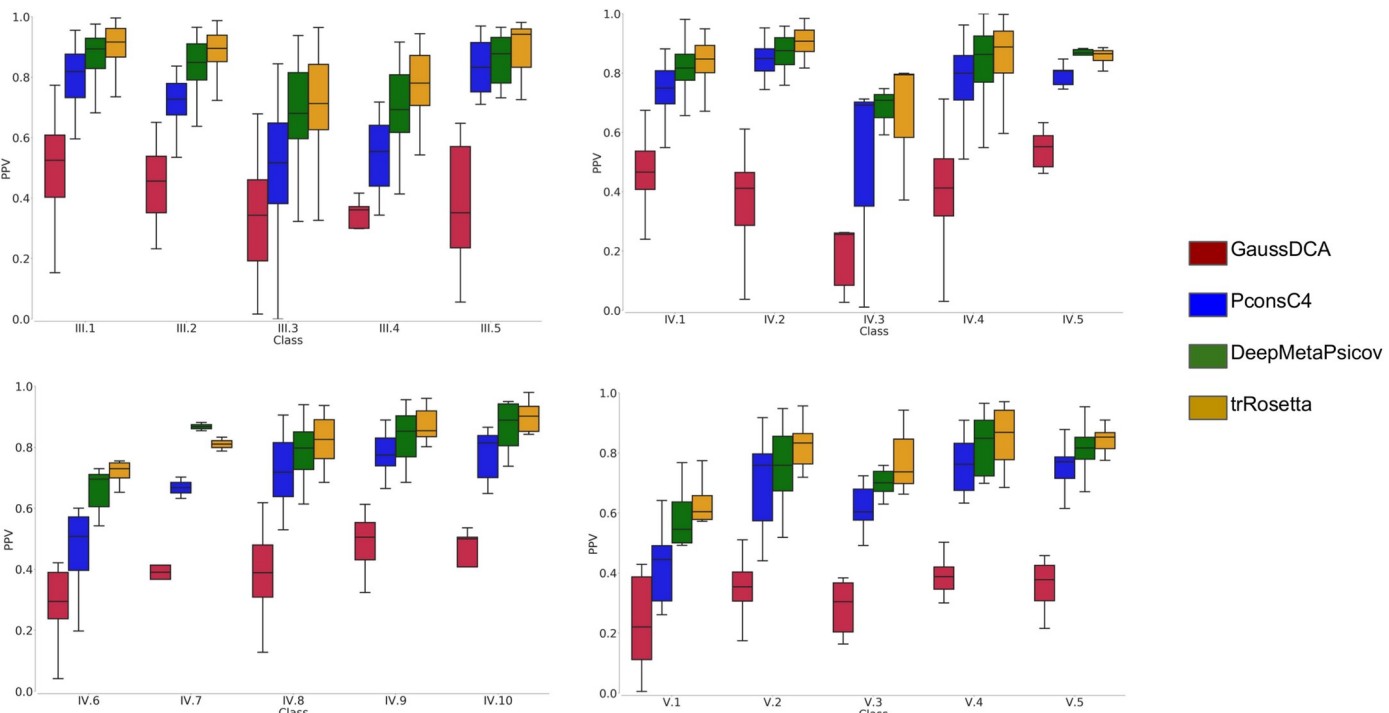

**Fig 2. The precision of contact predictions.** Positive Predictive Value (PPV) for the GaussDCA (red), Pconsc4 (Blue), DeepMetaPsicov (green), and trRosetta (orange).

repeat unit and a pair of repeats, obtaining in this way three datasets: i) a single unit datasets; ii) a double unit datasets; iii) complete repeat region datasets.

For all the three sets, multiple sequence alignments (MSA) and secondary structure predictions were generated. Subsequently using the MSA as input for trRosetta, PconsC4, DeepMetaPsicov, and GaussDCA contacts were predicted for each family. The performance of the contact predictions was then evaluated for each subclass separately. As expected, the most recent method, trRosetta outperforms an older deep learning method as PconsC4 and a simple DCA method as GaussDCA, but even ifcompared with the more recent, DeepMetaPsicov trRosetta shows a consistent improvement among all but two classes, Fig 2. In general for all the methods the predictions for the full-length regions give better results than when splitting the proteins into smaller units, Figs 3 and S1. In class V however, which is composed of entire domains, forming repeats of the *"beads on a string"* type, the splitting in units sometimes helps to reach better contact predictions for PconsC4, DeepMetaPsicov, and GaussDCA S1 Fig.

Here, it should be remembered that trRosetta, PconsC4 and DeepMetaPsicov, in addition to other information, use DCA predictions as an input and then learn to recognise specific patterns [18]. Therefore, artefacts present in the DCA predictions might propagate into these methods. In Fig 4, selected contact maps are shown as examples. The GaussDCA predictions contain periodic artefacts of wrong predictions (red dots) forming diagonal lines, occurring between equivalent positions in the repeat unit. PconsC4, DeepMetaPsicov, trRosetta appear efficient in removing the artefacts seen in GaussDCA. Here, it can be noted that there is only limited overlap between our repeat protein set and the training set of PconsC4 and DeepMetaPsicov, 25 out of 2856 and 29 out of 3456 proteins are identical respectively. Further, the accuracy for the shared proteins does not show a higher precision than the other proteins,

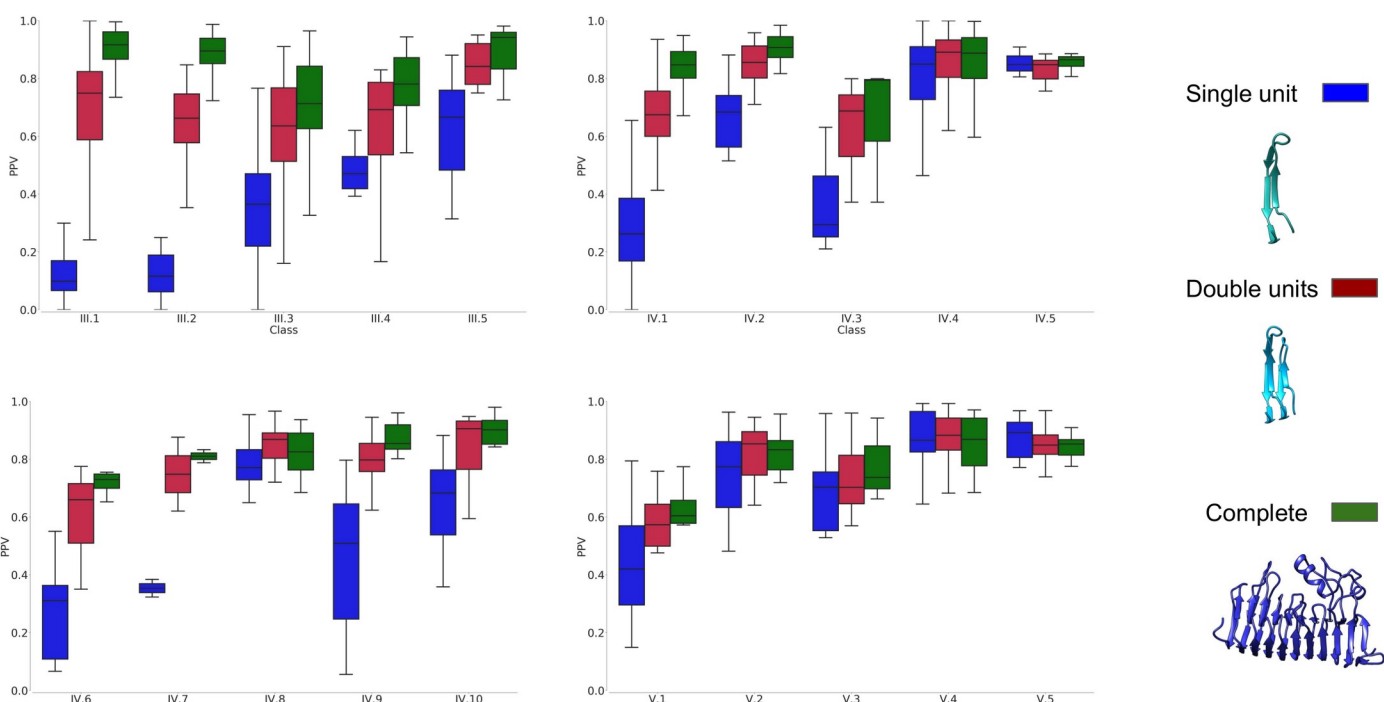

**Fig 3. The precision of contact predictions of trRosetta for the three datasets.** Results are shown for the three datasets, in blue the single unit dataset, in red the double units dataset, and in green complete region dataset.

S2 Fig. trRosetta instead has a much bigger training set of 15,051 proteins [19]. To our best knowledge, the IDs of the proteins are not available and in this case, we can not test the performance for the shared proteins. However, the high general consistency shown by trRosetta in our benchmark makes us confident that the results can be generalised, and that it is not strongly affected by a potential overlap with the training set.

It is well known that the prediction quality is directly correlated with the number of sequences in the starting MSA for DCA methods [18]. Here, this trend is also observed, with trRosetta always showing the best performance Fig 5.

## Differences among repeat classes in contacts prediction

Fig 2 shows variations in the fraction of correctly predicted contacts among different protein repeat classes and subclasses in all the methods. To clarify the origin of these differences, we investigated, more in-depth, the source of the predicted contacts. One central aspect that affects the difficulty of prediction is the pattern of the contacts [24]. In general, contacts that are parts of larger interaction areas or close in the sequence are predicted more accurately. Therefore, we compared the intra-unit and inter-unit contacts predicted by DeepMetaPsicov and trRosetta, Fig 6. Here, we obtained the number of predicted intra and inter-unit contacts from the PDB structures and selected the same number of predicted intra- and inter-units contacts. The PPV was finally calculated using the number of correctly predicted contacts divided by the number of contacts.

On average the intra-units contacts are predicted with higher accuracy than the inter-unit contacts in both DeepMetaPsicov and trRosetta, with trRosetta slightly over perform DeepMetaPsicov in both.

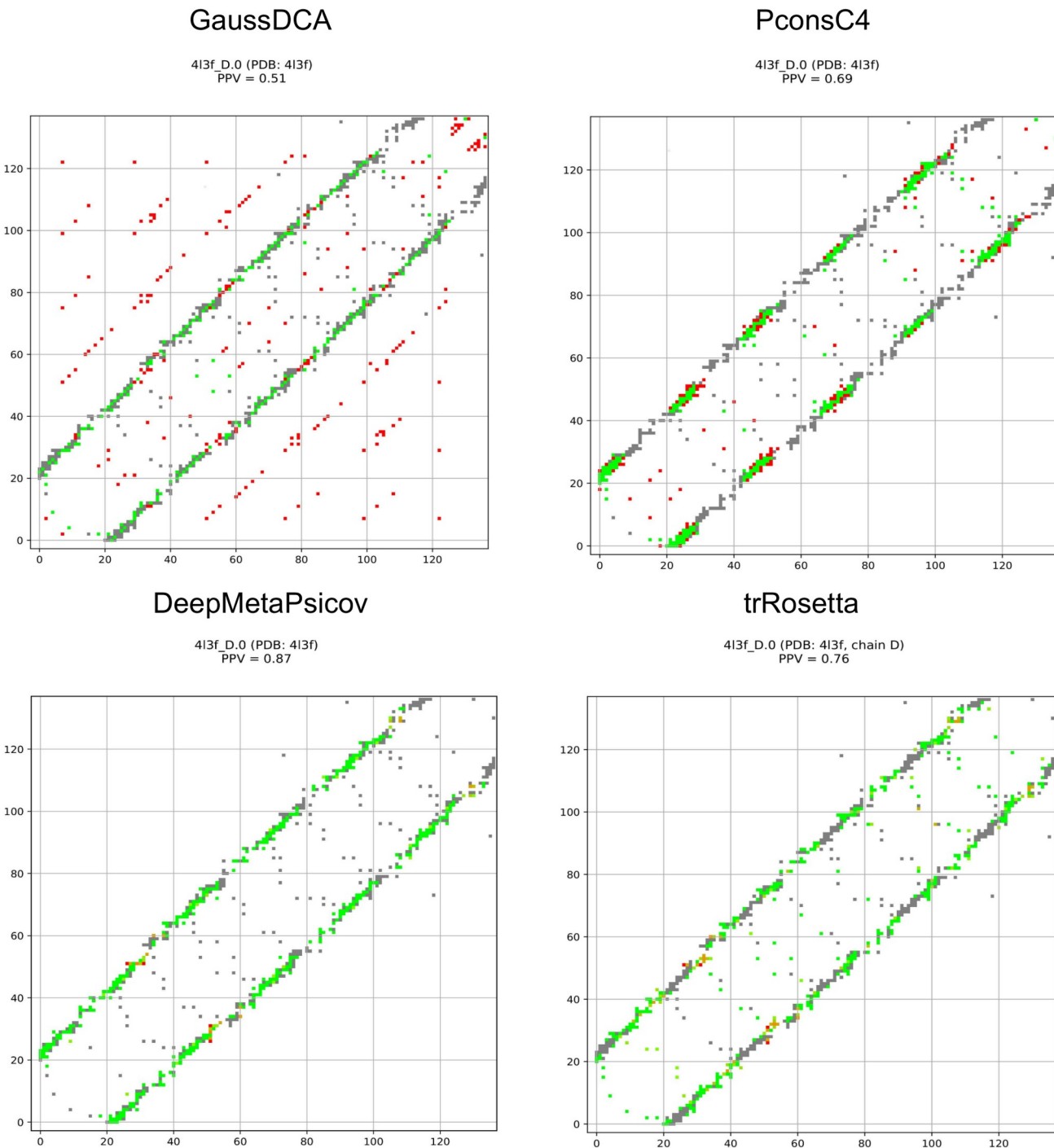

**Fig 4. GaussDCA, PconsC4, DeepMetaPsicov, and trRosetta contact maps.** Contact map for predictions obtained with GaussDCA, PconsC4, DeepMetaPsicov and trRosetta. In grey, the real contacts from the structure, in green, the corrected predicted contacts, and the falsely predicted contacts in red.

## Protein model generation

For PconsC4 and DeepMetaPsicov, protein models were generated using CONFOLD [25] using the contact predictions from the respective method and combining it with secondary

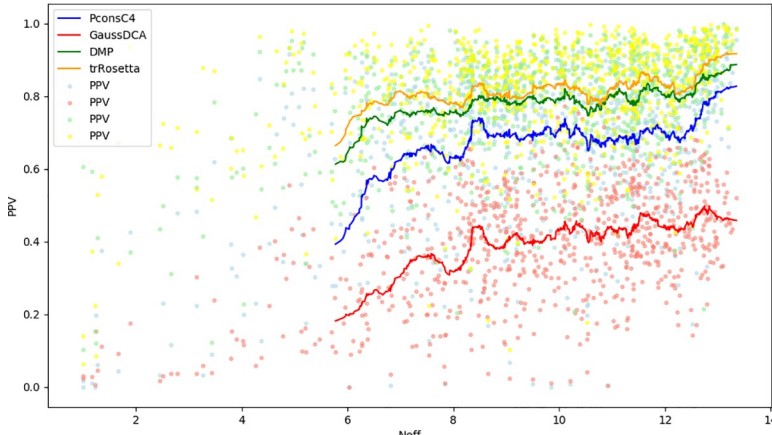

**Fig 5. The relation between Precision and the effective number of sequences in the MSA.** Positively Predicted Value for trRosetta in orange, GaussDCA in red, PconsC4 in Blue and DeepMetaPsicov in green on the Neff value (the effective number of sequences length weighted with the length of the protein). The single dots correspond to each protein in the datasets, and the line is the running average on (n = 50).

structure predictions from PSIPRED. For trRosetta instead, pyRosetta [26] was used for the protein model folding with the predicted distances and angles as input as described in Yang et al. [19]. Here, no secondary structure predictions were used.

In Fig 7, we compare the TM-score between the models of the corresponding PDB protein structure. Here, the trRosetta pipeline outperforms the other two methods in all classes but has to be noted that the use of distances and angles instead of contacts is the main responsible for the difference in performance with DeepMetaPsicov that when compared on the contacts prediction precision show slightly inferior performance. In total with trRosetta 732 models out of 815 (89.8%) are predicted with at least a TM-score of 0.5.

## Quality assessment of the models

To evaluate the quality of the models obtained by trRosetta, we compare the TM-scores of the models with the quality assessment scores from Pcons [22] and QmeanDisCo [23]. Due to the general high quality of the models both the quality assessment methods fail to rank a significant number of models properly, Fig 8A and 8B.

To improve the quality estimation, we developed a Random Forest Regression method using multiple inputs (Pcons, QmeanDisCo, protein length). Five-fold cross-validation was performed on the complete region dataset. The method obtained an average accuracy of 83.6%, and an average absolute error of 0.09 TM-score, see Fig 9A. The Random Forest Regression predicts the TM-score better than Pcons and QmeanDisCo alone, Fig 9B. We found that nine features were helpful for the prediction of the TM-score, S3 Fig. The most important features are the Pcons score, the local QmeanDisCo score, and protein length.

## Modelling of repeat protein families without known structures

We selected 48 PFAM repeat-families without resolved structure and fed them through the trRosetta structure prediction pipeline.

Among the models, 41 out of 48 (85%) are predicted with a TM-score higher than 0.5, Table 1. For twelve of these families, we could identify a template with a GMQE score [27] higher than 0.4 using Swissmodel [28]. In these cases, homology models were generated for comparison with the contact based models. We compared the similarity of the contact-based

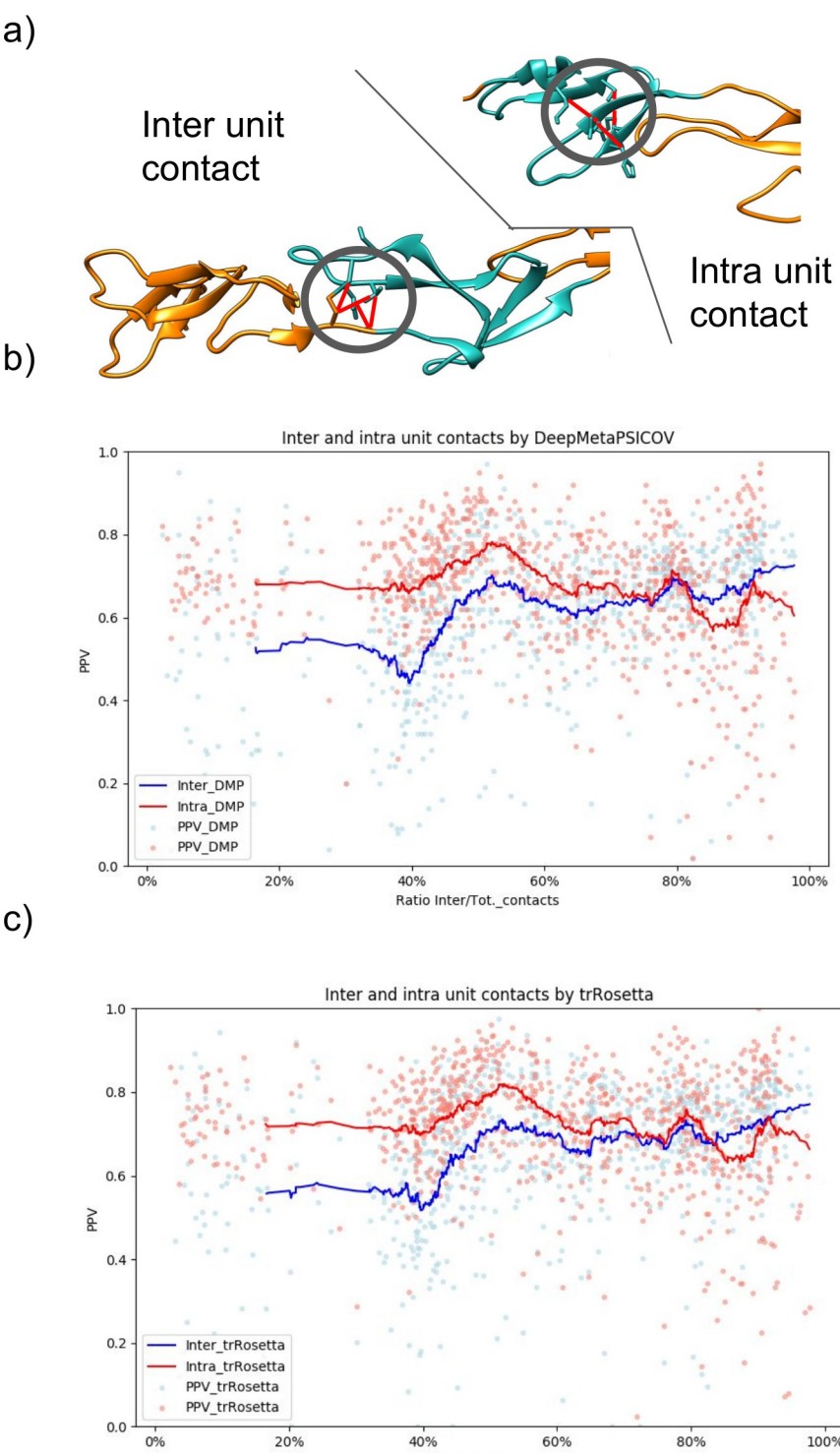

**Fig 6. Predicted contacts analysis.** a) Examples of inter- and intra- unit contacts. b) In red, the PPV for intra-units contacts in blue PPV for inter-units contacts predicted by DeepMetaPsicov. The lines are the respective running average of the PPV over the ratio of inter-unit contacts on the total of the protein contacts. c) In red, the PPV for intra-units contacts in blue PPV for inter-units contacts predicted by trRosetta. The lines are the respective running average of the PPV over the ratio of inter-unit contacts on the total of the protein contacts.

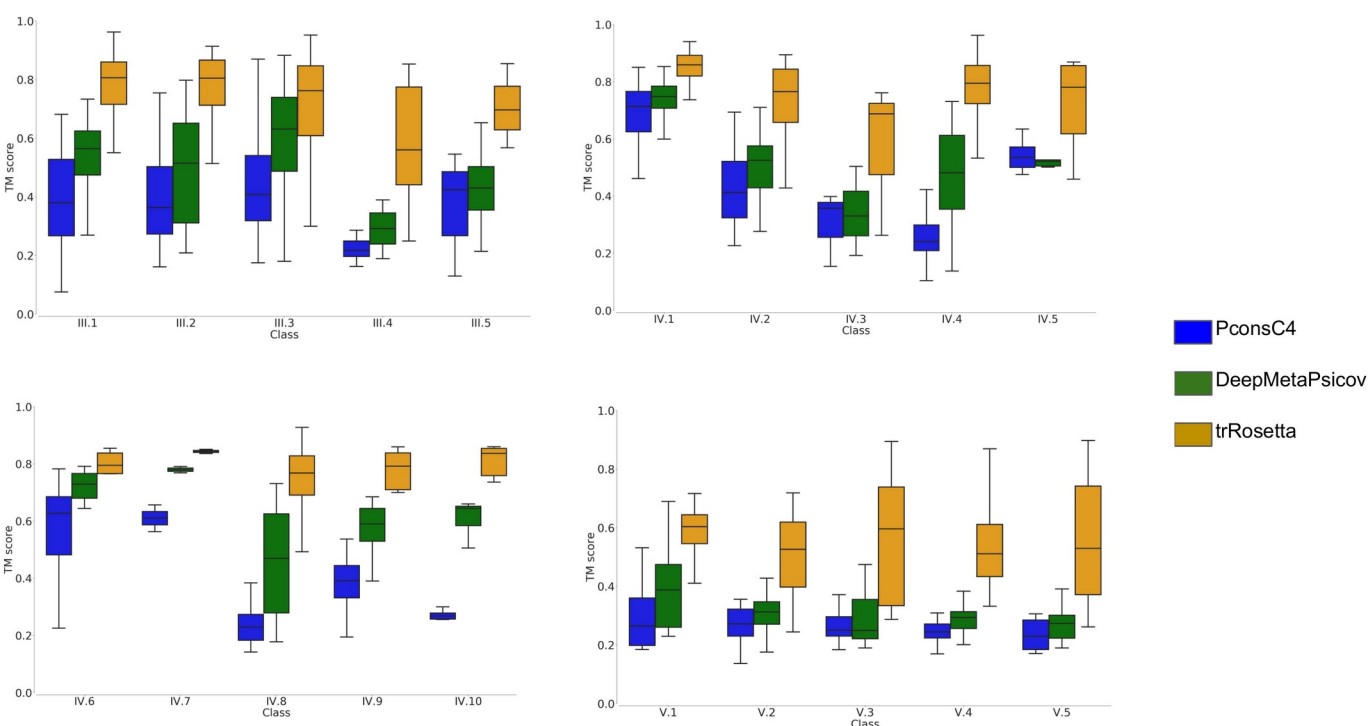

**Fig 7. Protein model quality.** TM-score for the subfamilies; Models from trRosetta in orange, PconsC4 in blue and DeepMetaPsicov in green.

and homology-based models with the predicted TM-score for the contact-based model. For four families (LVIVD, LRR_3, WD40_alt, LGFP) the models obtained by homology agree with the predicted TM-score, the difference between the TM-scores is below 0.1, i.e. the estimated TM-score agrees with what would be estimated if the homology model was identical to an experimental structure. However, for the other six families, there is an overestimation of the quality (RHS_repeat, DCAF15_WD40, DUF4116, Phage_fiber_2, *RTTN_N*, *MORN 2)* and for other two an underestimation (*DUF5122*, *FG-GAP_2)*, see Table 1.

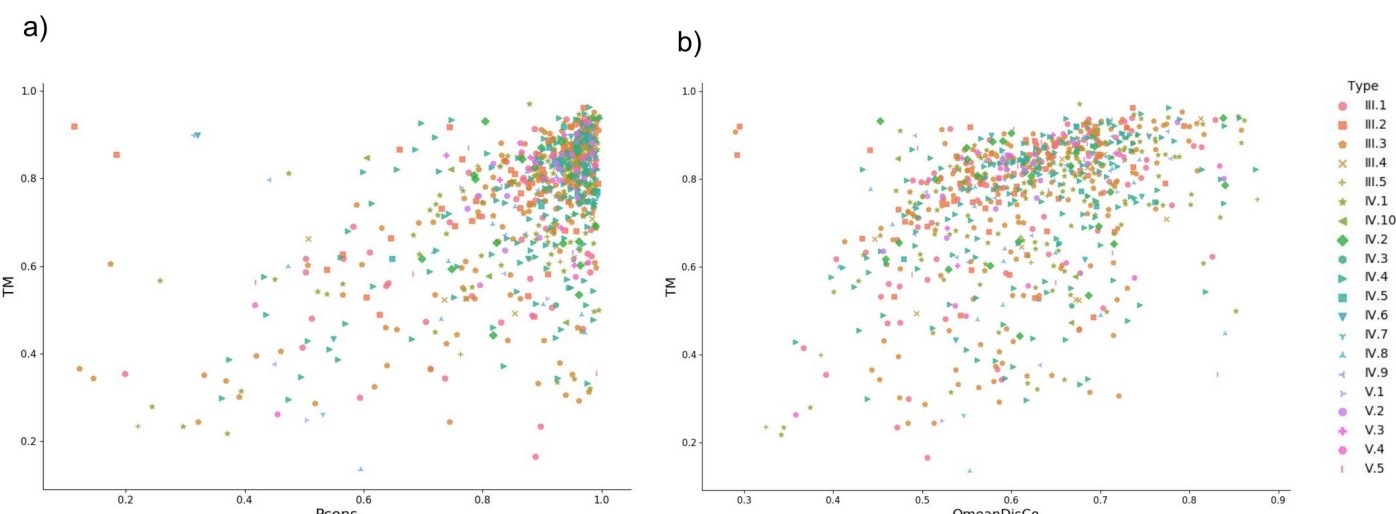

**Fig 8. TM-score versus QA methods.** a) TM-score versus Pcons-score for complete region models generated with trRosetta. b) TM-score versus QmeanDisCo score for full region models created from trRosetta contacts.

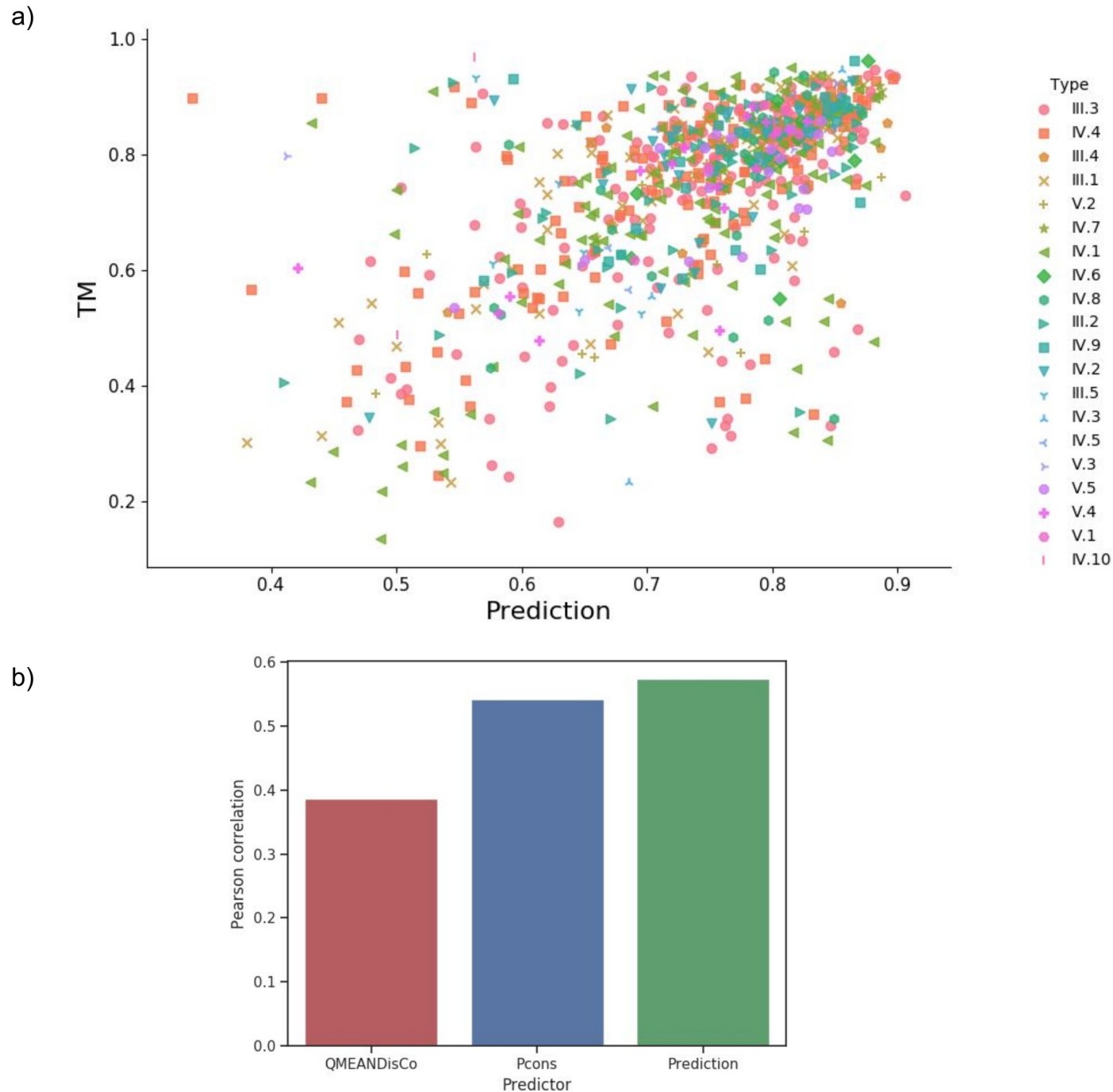

**Fig 9.** a) Real TM-score versus Random Forest Predicted TM-score for complete region models generated with trRosetta. b) Pearson correlation coefficient between the TM-score and the QA methods.

Fig 10 shows the overlap between the contact and the homology models. Two trRosetta models differ significantly from the homology models: Phage_fiber_2 (PF03406) where the template as a partially disordered extended structure while the trRosetta model is packed and DUF4116 (PF13475) in which the trRosetta model is folded as an α-solenoid while the homology model in a longer helical bundle.

**Table 1. The models of the PFAM families with predicted TM-score.** In the columns: the family name, the PFAM ID, the Uniprot ID of the sequence used for the modelling, the predicted TM-score, the best template PDB ID, the Swismodel GMQE score, the identity between the target/template alignment, the TM-score between the contact-based model and the homology model.

| PFAM family | PFAM ID | Representative protein Uniprot ID | TM prediction | Template PDB ID | GMQE | Identity | TM-score between The contact model and Homology model |
|---|---|---|---|---|---|---|---|
| Nebulin | PF00880 | A0A094KVK3 | 0.483185 | 6b40_A | 0.12 | 18.60% | - |
| SWM_repeat | PF13753 | A0A0B0HSH2 | 0.45394 | 2ra1_A | 0.21 | 20.72% | - |
| Plasmod_MYXSPDY | PF07981 | A0A0L7M9B8 | 0.592519 | - | - | - | - |
| C_tripleX | PF02363 | A0A1A9WU23 | 0.440582 | 4xbm_A | 0.28 | 29.08% | - |
| RHS_repeat | PF05593 | A0A1G0MXS8 | 0.789065 | *6fay_A* | *0.7* | *28.10%* | *0.51* |
| *Plasmodium repeat MYXSPDY* | PF00839 | A0A1I7SWM5 | 0.585336 | | 0.02 | 9.09% | - |
| LVIVD | PF08309 | A0A1V1NWB1 | 0.793165 | 4jsn_B | 0.5 | 12.50% | 0.78 |
| *DUF5122* | PF17164 | A0A1Z4C3E9 | 0.685674 | *2ymu_A* | *0.46* | *19.50%* | 0.82 |
| Bacterial tandem repeat domain | PF17660 | A0A252E8A5 | 0.74885 | *4qp0_A* | *0.17* | *13.70%* | - |
| SprB | PF13573 | A0A257INW4 | 0.602068 | 2c26_A | 0.31 | 16.36% | |
| Lustrin_cystein | PF14625 | A0A2A2LSA2 | 0.521748 | 6nan_A | 0.13 | 20.63% | - |
| *SPW* | PF03779 | A0A2A3HD64 | 0.687938 | - | - | - | - |
| Chlorovi_GP_rpt | PF06598 | A7RAI0 | 0.627261 | - | - | - | - |
| CRAM_rpt | PF07016 | A7S4G3 | 0.667426 | 4aea9_A | 0.39 | 17.65% | - |
| Dicty_CTDC | PF00526 | D3BR65 | 0.566775 | 4u8u_N | 0.17 | 19.23% | - |
| RtxA | PF07634 | D3UXB8 | 0.744867 | 5vgz_A | 0.03 | 20.51% | - |
| YTV | PF07639 | D5SU36 | 0.591834 | - | | | |
| LRR_3 | PF07725 | D7MCA5 | 0.697779 | 2omx_A | 0.59 | 18.97% | 0.71 |
| Ice_nucleation | PF00818 | F3GDU0 | 0.714233 | - | - | - | - |
| LSPR | PF06049 | G1RYA9 | 0.571161 | - | - | - | - |
| UCH-protein repeats | PF13446 | G1XIQ8 | 0.611722 | 2h5x_C | 0.08 | 22.45% | - |
| WD40_alt | PF00400 | G3VIY2 | 0.718139 | *5obm_A* | *0.59* | 22,78% | 0.71 |
| Lipoprotein_15 | PF03640 | I3BT02 | 0.449038 | 4yx7_A | 0.02 | 16% | |
| SSURE | PF11966 | J1S4N0 | 0.509385 | - | - | - | - |
| LGFP | PF08310 | L8TNF3 | 0.779499 | 6sx4_A | 0.76 | 32.21% | 0.85 |
| zf-C2H2_3rep | PF18868 | O64827 | 0.588789 | 1z9v_A | 0.04 | 12.12% | |
| DCAF15_WD40 | PF14939 | Q29AL9 | 0.677563 | 6pai_B | 0.67 | 29.69% | 0.56 |
| SVS_QK | PF10578 | Q6P6X2 | 0.605577 | - | - | - | - |
| DUF2963 | PF11178 | Q6YQH3 | 0.804535 | 5e9t_D | 0.32 | 24.24% | |
| Plasmo_rep | PF12135 | Q7RTC2 | 0.637834 | 4nee_C | 0.07 | 23.08% | |
| *Curlin* | PF07012 | Q8EIH3 | 0.669206 | - | - | - | - |
| *MORN 2* | PF07661 | Q8RH85 | 0.71293 | *1h3i_A* | *0.68* | *27.87%* | 0.52 |
| OGFr_III | PF04680 | Q9NZT2 | 0.679499 | 5xme_A | 0.21 | 16.09% | |
| HNH_repeat | PF18780 | R2SEH8 | 0.648563 | 2xsj_C | 0.24 | 14.67% | |
| ChW | PF07538 | R5P8A5 | 0.674395 | - | - | - | - |
| WG_beta_rep | PF14903 | R6YH89 | 0.681589 | 2ki4_A | 0.06 | 7.41% | |
| DUF4116 | PF13475 | R7MCC4 | 0.688163 | 5lu2_A | 0.41 | 7.81% | 0.33 |
| PHINT_rpt | PF14882 | S6TLB9 | 0.460735 | 5lnk_Q | 0.03 | 13.51% | |
| Chlam_PMP | PF02415 | S7J9T7 | 0.674383 | 2m7o_A | 0.13 | 25% | |
| Xin | PF08043 | T0NQR8 | 0.680028 | 1ixv_A | 0.1 | 8.33% | |
| WXXGXW | PF12779 | U2FCE1 | 0.731013 | - | - | - | - |
| *SBBP* | PF06739 | U5QIU9 | 0.724106 | *6i3b_A* | *0.38* | *24.45%* | |
| Ish1 | PF10281 | U7Q0S5 | 0.433415 | 1jjr_A | 0.1 | 15.73% | |

*(Continued)*

**Table 1.** (Continued)

| PFAM family | PFAM ID | Representative protein Uniprot ID | TM prediction | Template PDB ID | GMQE | Identity | TM-score between The contact model and Homology model |
|---|---|---|---|---|---|---|---|
| Phage_fiber_2 | PF03406 | V5CQL0 | 0.540125 | 5iv5_A | 0.58 | 22.43% | 0.21 |
| FG-GAP_2 | PF14312 | W4LGN0 | 0.640251 | 5ffg_A | 0.56 | 23.04% | 0.82 |
| CXCXC | PF03128 | W5N853 | 0.865153 | 1vgh_A | 0.36 | 26.47% | |
| RTTN_N | PF14726 | W5P499 | 0.6874 | 4plr_A | 0.58 | 16.28% | 0.58 |
| WDCP | PF15390 | W5Q8K9 | 0.427924 | 5nnz_A | 0.09 | 12.58% | |

Other 36 families do not have suitable templates, and, therefore, we cannot compare their trRosetta models with a homology-based model. However, the quality assessment shows high scores for the vast majority of the models.

Here we describe a few interesting models in more details, and all the models are available at https://figshare.com/articles/dataset/Repeats_Proteins_contact_prediction_based_modelling_datasets/9995618. We do encourage others to investigate the other models in details.

## SPW family (PF03779)

According to the PFAM database [29], the SPW family is present in Bacteria and Archaea, and each protein consists of one or two repeat units. Some members also contain an additional domain, either a Vitamin K epoxide reductase (PF07884) or a NAD-dependent epimerase/dehydratase (PF01370). Each repeat unit is formed by two transmembrane alpha-helices and is characterised by an SPW motif [30]. According to our model, the repeated motif is buried in the membrane symmetrically located close to the extracellular side, Fig 11B. PFAM architectures show many proteins with only a single SPW motif however a more careful analysis of these sequences shows that in many cases they contain a second degenerate SPW unit with the proline residue conserved (S4 Fig).

The Tryptophan is on the outer side of the protein, facing the bilayer, while the proline is on the inner side of the protein, promoting the formation of a kink in the transmembrane helix [31]. The protein contains a ser-pro motif, rare among TM-proteins and most likely increases the bending effect of proline significantly due to their hydrogen bond pattern [32].

## Curlin repeats family (PF07012)

Here, the trRosetta model has a higher predicted TM score (Table 1) and agrees better with information from the available literature [37]. Curlin is predicted to have a β-solenoid structure, see Fig 11C. DeBenedictis et al. presented ab-initio models for two members of the Curlin repeat family, CsgA and CsgB [37]. The structure of their best models is visually in agreement with our model (a direct comparison is difficult as the coordinates are not available for their models). Our model is also in agreement with the partial structure of the repeat units of CsgA published by Perov et al. [38]. This model contains two parallel β-sheets with individual units situated perpendicular to the fibril axis (corresponding PDB IDs are 6G8C, 6G8D, 6G8E).

## UCH-protein (PF13446)

Our model (Fig 11D) suggests that this repeat region is a Class V.1 α-beads, with four helical domains separated by a flexible linker.

UCH-protein repeats family is a repeat domain found in Ubiquitin carboxyl-terminal hydrolase. Despite UCH-proteins being widespread among eukaryotes, the repeated domain is

## Overlap Contact-model vs Homology model

trRosetta Models Homology Model

RHS_repeat (PF05593) LVIVD (PF08309)

TMscore 0.51 TMscore 0.78

*DUF5122*(PF17164) LRR_3 (D7MCA5)

TMscore 0.82 TMscore 0.71

WD40_alt (PF00400) LGFP (PF08310)

TMscore 0.71 TMscore 0.85

*MORN 2* (PF07661) DCAF15_WD40 (PF14939)

TMscore 0.52 TMscore 0.56

*FG-GAP_2* (PF14312) Phage_fiber_2 (PF03406)

TMscore 0.51 TMscore 0.221

DUF4116 (PF13475) *RTTN_N* (PF14726)

TMscore 0.33 TMscore 0.58

**Fig 10. Comparison between the contact-based model and homology modelling.** The superposition between the contact-based model (red) and the homology model (blue) and respective TM-score.

present only in yeasts in a variable number of units. According to PFAM [29], the UCH-protein repeats could be involved in the formation of a complex of UCH with Rsp5 and Rup1.

### Xin repeat (PF08043)

Xin repeats is a motif with a variable number of units, known for binding and stabilising F-actin [33]. In mouse and chicken is located in the adherens junction complex [33]. In humans Xin-repeat proteins are involved in the developmental and adaptive remodelling of the actin cytoskeleton [34] behaving as a scaffold protein showing multiple interacting partners: I) interact with the EVH1 domain of Mena/VASP/EVL [34]. II) Interact with the SH3 domain of Nebuline and Nebulette, despite the binding site is located in a disordered region [35] III) interact with Aciculin [36].

In our model Fig 11E, Xin-repeats result folded as an α-solenoid. This clarifies the fold of Xin-repeats proteins formed by an α-solenoid N-terminus and a long disordered C-terminal region.

## Conclusion

Here, we performed a comprehensive coevolution analysis on repeat protein families, and we show that trRosetta contact-predictions method overcomes the traditional difficulties of previews Deep Learning and DCA methods for this class of proteins. We investigated the modelling of repeat units, and we developed a novel quality assessment method for repats proteins. Finally, we tested the pipeline on PFAM families without protein structures showing its usefulness in providing new structural information.

This paper summarises the extraordinary improvement of the structure prediction method in the past few years and shows that it is now possible to predict the structure of 85% of PFAM repeat families satisfactorily.

## Materials and methods

### Datasets generation

The repeat protein dataset was generated starting from the 3585 reviewed entries in RepeatsDB [14,39]. The proteins of class I and II were removed, and then the dataset was homology reduced using CD-HIT [40] at 40% identity resulting in 815 repeat regions. From this "complete region dataset" two other datasets were generated. First, a "single unit" dataset with one repeat unit from each family, and secondly a "double unit" dataset with two. In the two derived datasets, the representative units were selected, avoiding or at least minimising, the presence of insertions.

The non-resolved repeats protein family dataset was generated, collecting all the repeat proteins families with missing structural information present in PFAM [29] as of May 2019 and removing domains with a significant overlap with the disorder prediction. It results in 48 protein families. The representative sequence for each family of repeat was chosen for matching these criteria: 1) select the most common architecture; 2) Include when possible at least three repeat units.

### Multiple sequence alignment (MSA)

The multiple sequence alignments (MSA) were carried out using HHblits [41] using an E-value cutoff of 0.001 against the Uniclust30_2017_04 database [42]. The number of effective

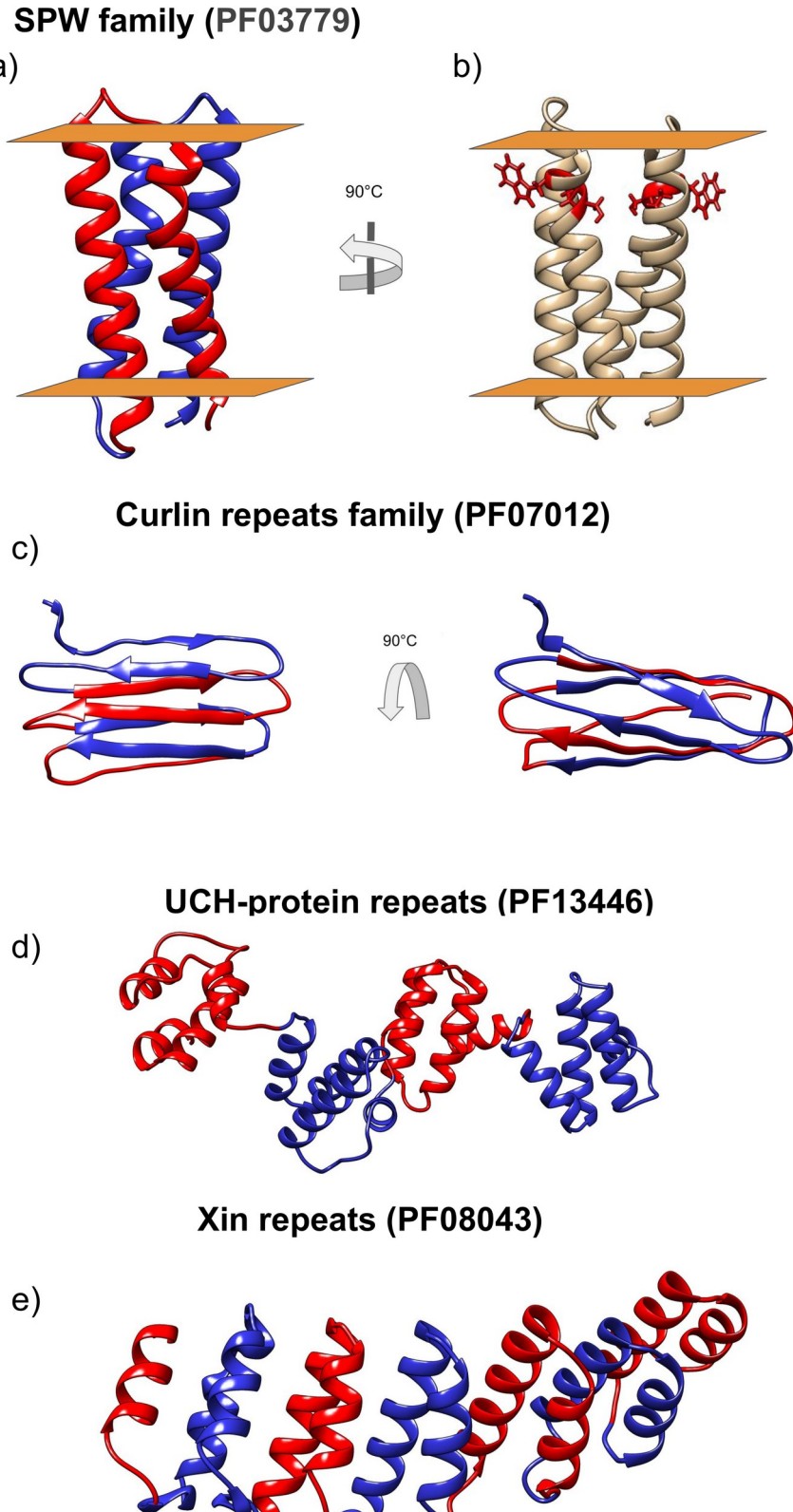

**Fig 11. Selected models.** The different protein units are coloured in red and blue. a) SPW, b) SPW in red the "SPW" motif c) Curlin d) UCH-protein are shown and e) Xin repeat.

sequences of the alignment, expressed as Neff-score, was calculated by HHblits and used for subsequent analysis. More detail about the Neff calculation can be found at https://github.com/soedinglab/hh-suite/wiki [41].

### Contact prediction and models generation

For DeepMetaPsicov and PconsC4 the protein models were generated following the Pcons-Fold2 protocol [43]. The secondary structure of the repeat regions was predicted by PSIpred [44]. Protein contacts were predicted using DeepMetaPsicov [45], or PconsC4 [18] and together with the secondary structure predictions used as input to Confold [25]. The modelling used the top scoring 1.5 L contacts (where L is the length of the modelled regions).

The Rosetta models were obtained running trRosetta locally [19] and use the predicted distances and angles as input for pyRosetta [26].

### Contacts analysis

A protein contact was defined as two residues having a beta carbon distance equal to or lower than 8Å in the PDB structure and farther than five residues in the sequence. Using this definition, we assess the number of correctly predicted contacts the Positively Predicted Value (PPV) taking into account the top-scoring 1.5 L contacts.

Since trRosetta predicted distances instead of contacts between the residues we sum the probabilities for the distance bin equal or shorter than 8 Å as in Greener et al. [20] in order to compare them with the contacts predicted with the other methods.

In the intra/inter-unit contacts analysis, the predicted contacts of each protein were divided into i) intra-unit contacts, if between residues inside the same unit; ii) inter-units if the residues are in different repeat units. The units mapping was taken from the RepeatsDB database [14]. In this analysis, we calculate the number of intra- and inter-unit contacts in the PDB structure, and then we selected the same number of predicted intra- and inter-units contacts. The PPV was then calculated as the fraction of correct predictions.

### Template search and homology modelling

The template search and the homology models were generated from the representative sequences using the default options from Swissmodel [28].

### Protein models analysis

The model quality, expressed in TM-score, was assessed using a random forest regression model using the python module Sklearn. The random forest regression was optimized to include 240 estimators and a maximum depth of 60. The models from the trRosetta "complete region" benchmark set were used as a training set. The label of the training set was the TM-score of each model [46]. To ensure that the protein structure and the model were aligned correctly, the TMalign option -I was used, providing a local alignment of the two sequences.

For training, five cross-validation sets were generated. Several inputs were used for the random forest, described briefly below and in Table 1. The Confold and QmeanDisco inputs were obtained from analysing the first ranked model. Pcons was run using the option -d using all the models in the stage2 folder generated by Confold. Among the different sets of features tried, we select nine features that all improve the prediction of the random forest regression, see S3 Fig.

## Supporting information

**S1 Fig. The precision of contact predictions.** Positive Predictive Value (PPV) for the GaussDCA (red), Pconsc4 (Blue), and DeepMetaPsicov (green). For all three methods results are shown for the three datasets, in light colour the single unit dataset, in intermediate colour the double units dataset, and in the darker colour the complete region dataset.
(TIF)

**S2 Fig. Performance expressed in PPV for the set of proteins contained in the training set of the contact predictions methods.** In light green, the Deep Meta Psicov (DMP) average precision for the proteins of the benchmark set not overlapping with the DMP training set, in dark green the DMP average precision for the proteins benchmark set present in the DMP training set, in light blue the PconsC4 average precision for the proteins of the benchmark set not overlapping with the PconsC4 training set, in dark blue the PconsC4 average precision for the proteins benchmark set present as well in the PconsC4 training set.
(TIF)

**S3 Fig. Random Forest model features importance.** The features used in the random forest model are listed according to their relative importance.
(TIF)

**S4 Fig. Amino Acid frequency of the single domain architecture sequences.** From the logo is possible to recognize two SPW domains, one of them degenerated (in particular the first Serine in the second motif) that is not recognized by PFAM.
(TIFF)

## Author Contributions

**Conceptualization:** Claudio Bassot, Arne Elofsson.

**Data curation:** Claudio Bassot.

**Formal analysis:** Claudio Bassot.

**Funding acquisition:** Arne Elofsson.

**Investigation:** Claudio Bassot.

**Methodology:** Claudio Bassot.

**Project administration:** Arne Elofsson.

**Resources:** Arne Elofsson.

**Software:** Claudio Bassot.

**Supervision:** Arne Elofsson.

**Validation:** Claudio Bassot, Arne Elofsson.

**Visualization:** Claudio Bassot.

**Writing – original draft:** Claudio Bassot, Arne Elofsson.

**Writing – review & editing:** Claudio Bassot, Arne Elofsson.

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
