## [Decision Letter · Decision Letter 0]

7 Oct 2020

Dear Arne,

Thank you very much for submitting your manuscript "Accurate contact-based modelling of repeat proteins predicts the structure of new repeats protein families." for consideration at PLOS Computational Biology.

As with all papers reviewed by the journal, your manuscript was reviewed by members of the editorial board and by several independent reviewers. In light of the reviews (below this email), we would like to invite the resubmission of a significantly-revised version that takes into account the reviewers' comments.

Two reviewers felt that the manuscript did not contain the amount of new methodological innovation expected of a PLoS Computational Biology article. Reviewer 2 suggested to include distances predicted by RaptorX-Contact to the evaluation. As such, the paper would earn its merits more as a benchmarking study, which seems to make sense, given that you have already included constraints from two different contact prediction methods. (If there are additional distance/contact predictors that can be included, it makes sense to include these too.) If you can add this analysis, we would be interested to receive a revision. We are sorry to not be able to be more positive at this point. 

We cannot make any decision about publication until we have seen the revised manuscript and your response to the reviewers' comments. Your revised manuscript is also likely to be sent to reviewers for further evaluation.

Sincerely,

Johannes Soeding, Ph.D.

Guest Editor

PLOS Computational Biology

Nir Ben-Tal

Deputy Editor

PLOS Computational Biology

Reviewer's Responses to Questions

**Comments to the Authors:**

Reviewer #1: The paper deals with an interesting and novel aspect of protein feature predictions. The authors took into considerations all the major criticisms raised by reviewers and edited the paper accordingly. The problem of redundancy is now correctly described.

Reviewer #2: The authors did some modifications of the manuscript by testing the contacts predicted by DeepMetaPSICOV and developing a random forest regression method to predict the quality of a 3D model. However, the authors did not test folding of distance (and orientation information used by trRosetta). It is unclear why the authors did not test the distance (and orientation) information predicted by RaptorX-Contact simply because it is a web server. In fact, it is much more convenient to use a web server than a standalone software package. A hassle-free strategy is to submit the test proteins to the RaptorX-Contact web server and then evaluate the contact/distance prediction and 3D models returned by the server. Overall, this manuscript represents a simple application of existing (but not very cutting-edge) techniques to an interesting problem. As a proof of concept, it may be sufficient, but it may not represent what we can do on the modeling of repeat proteins using existing techniques and tools.

Reviewer #3: I find that the paper does not features works of 'exceptional significance', but a specific evaluation of distinct methods to asses a specific technical question. The exceptional significance would be if the paper fully describe and asses novel structures for repeat proteins based on these findings, but I that this is lacking in the manuscript.

**Have all data underlying the figures and results presented in the manuscript been provided?**

Reviewer #1: Yes

Reviewer #2: None

Reviewer #3: None

PLOS authors have the option to publish the peer review history of their article (what does this mean?). If published, this will include your full peer review and any attached files.

Reviewer #1: No

Reviewer #2: No

Reviewer #3: No
---

## [Decision Letter · Decision Letter 1]

15 Feb 2021

Dear Arne,

We are pleased to inform you that your manuscript 'Accurate contact-based modelling of repeat proteins predicts the structure of new repeats protein families.' has been provisionally accepted for publication in PLOS Computational Biology.

Best regards,

Johannes Soeding, Ph.D.

Guest Editor

PLOS Computational Biology

Nir Ben-Tal

Deputy Editor

PLOS Computational Biology

Reviewer's Responses to Questions

**Comments to the Authors:**

Reviewer #2: The authors have applied trRosetta to their proteins and improved the results, and I think now the work is ready for publication.

By the way, in CASP14 Baker's trRosetta worked better than RaptorX mainly because the former used very large metagenome databases to generate MSAs while the latter does not, which is very important in CASP14 since it has quite a few test targets (from the same complex) that need metagenome data. Nevertheless the authors do not seem to use any metagenome data in this manuscript. The whole RaptorX package is also publicly available at https://github.com/j3xugit/RaptorX-3DModeling for local installation.

Reviewer #3: ok

**Have all data underlying the figures and results presented in the manuscript been provided?**

Reviewer #2: None

Reviewer #3: Yes

PLOS authors have the option to publish the peer review history of their article (what does this mean?). If published, this will include your full peer review and any attached files.

Reviewer #2: No

Reviewer #3: No

---

## [Editor Report · Acceptance letter]

31 Mar 2021

PCOMPBIOL-D-20-01386R1 

Accurate contact-based modelling of repeat proteins predicts the structure of new repeats protein families.

Dear Dr Elofsson,

I am pleased to inform you that your manuscript has been formally accepted for publication in PLOS Computational Biology. Your manuscript is now with our production department and you will be notified of the publication date in due course.

With kind regards,

Katalin Szabo
